# Interventions for Promoting Meconium Passage in Very Preterm Infants—A Survey of Current Practice at Tertiary Neonatal Centers in Germany

**DOI:** 10.3390/children9081122

**Published:** 2022-07-27

**Authors:** Maximilian Gross, Helmut Hummler, Bianca Haase, Mirja Quante, Cornelia Wiechers, Christian F. Poets

**Affiliations:** Department of Neonatology, University Children’s Hospital Tuebingen, 72076 Tuebingen, Germany; helmut.hummler@med.uni-tuebingen.de (H.H.); bianca.haase@med.uni-tuebingen.de (B.H.); mirja.quante@med.uni-tuebingen.de (M.Q.); cornelia.wiechers@med.uni-tuebingen.de (C.W.); christian-f.poets@med.uni-tuebingen.de (C.F.P.)

**Keywords:** meconium, enema, contrast agent, polyethylene glycol, acetylcysteine, necrotizing enterocolitis, survey

## Abstract

Meconium passage is often delayed in preterm infants. Faster meconium passage appears to shorten the time to full enteral feeds, while severely delayed meconium passage may indicate meconium obstruction. Neonatologists often intervene to promote meconium passage, assuming that benefits outweigh potential risks such as necrotizing enterocolitis (NEC). We performed an anonymous online survey on different approaches to facilitate meconium passage among tertiary neonatal intensive care units (NICUs) in Germany between February 2022 and April 2022. We collected information on enteral nutrition, gastrointestinal complications, and interventions to promote meconium passage. We received 102 completed questionnaires (response rate 64.6%). All responders used interventions to promote meconium passage, including enemas (92.0%), orally applied contrast agents (61.8%), polyethylene glycol (PEG) (46.1%), acetylcysteine (19.6%), glycerin suppositories (11.0%), and maltodextrin (8.8%). There was substantial heterogeneity among NICUs regarding frequency, composition, and mode of administration. We found no differences in NEC incidence between users and nonusers of glycerin enemas, high or low osmolar contrast agents, or PEG. There is wide variability in interventions used to promote meconium passage in German NICUs, with little or no evidence for their efficacy and safety. Within this study design, we could not identify an increased risk of NEC with any intervention reported.

## 1. Introduction

While meconium passage is often delayed and prolonged in very preterm infants [1], early meconium evacuation is associated with a shortened time to full enteral feeding, reduced central venous line use and hospital stay [2,3,4,5]. Therefore, various interventions have been studied to promote meconium evacuation in preterm infants, including enemas, suppositories, rectal stimulation, and enteral application of a contrast agent [5,6,7,8,9]. Although some of these interventions shortened the time to full enteral feeds [5,10], the overall evidence for routinely promoting meconium evacuation to facilitate feeding tolerance and accelerate meconium passage in preterm infants is limited [6,9,11]. In addition, conflicting data exist regarding the association of enemas and enteral administration of contrast agents with necrotizing enterocolitis (NEC) [5,8,12]. Thus, neonatologists’ approaches are quite variable and may include watchful waiting, early or late interventions (starting interventions routinely during the first postnatal days vs. only if no meconium is passed within a certain time after birth) or a use of interventions only in symptomatic infants, i.e., with feeding intolerance or abdominal distension.

To assess current approaches to meconium evacuation in very preterm infants and their potential association with NEC, we conducted a national survey in all tertiary neonatal intensive care units (NICUs) in Germany. Due to the lack of evidence, we expected to observe wide variability in individual practices.

## 2. Materials and Methods

### 2.1. Study Design

We designed this national cross-sectional survey according to the CROSS guidelines for survey studies [13]. An online questionnaire (Appendix A) was created using SoSci Survey (SoSci Survey GmbH, Munich, Germany) and circulated via e-mail among tertiary NICUs in Germany between February 2022 and April 2022. Included were all tertiary NICUs in Germany. The exclusion criterion was an incomplete response to the questionnaire. This study was registered at the German Register of Clinical Trials (trial no. DRKS00028274) and was approved by the ethics committee of Tuebingen University Hospital (application no. 012/2022BO2). Since this was an anonymous data analysis, consent was given by voluntary participation in the online survey.

### 2.2. Questionnaire and Study Population

The anonymous questionnaire consisted of up to 42 questions divided into five sections: (I) NICU’s and neonatologists’ characteristics; (II) approach to enteral nutrition; (III) gastrointestinal complications, i.e., annual NEC rate based on the last five years in the unit; (IV) use of systemic steroids and treatment of patent ductus arteriosus; and (V) institutional approach to interventions used for meconium mobilization. In Section V, participants could choose between different interventions to promote meconium passage. In case of watchful waiting as the general institutional approach, the survey was finished; if participants affirmed the general use of interventions to promote meconium evacuation, they were prompted to provide more details on the actual method used in their NICU. Response options were single, multiple-choice or text input, respectively. Six experienced neonatologists at our institution created the questionnaire, and the final set of questions used in the survey was circulated after repeated pre-testing.

The target population were the respective medical directors of all tertiary NICUs in Germany, listed in a yearly updated national information database for quality assurance in very-low-birth-weight infants available to the public [14]. Tertiary NICUs in Germany provide care for preterm infants <29 weeks of gestation or with an estimated birth weight <1250 g. As of February 2022, 167 tertiary NICUs were registered at the above information portal. We identified two double entries, resulting in 165 NICUs; seven medical directors supervised two NICUs. We contacted the 158 respective medical directors via e-mail. They received an individual link to the questionnaire which allowed them to fill in the questionnaire only once to avoid double entries. We sent out two e-mail reminders after four and seven weeks, respectively.

### 2.3. Statistical Analysis

We imported survey data from the SoSci Survey database of the Statistical Package for Social Sciences Version 27 (SPSS, IBM Corp., Armonk, NY, USA) for analysis and descriptive data reporting. Data are presented as total response frequency with percentages in parenthesis or as median (minimum and maximum). We evaluated differences between groups using the Mann-Whitney U-test in non-normally distributed numerical factors. The statistical significance was set at a *p*-value of < 0.05.

## 3. Results

A total of 105 NICUs completed our online survey, corresponding to a response rate of 66.5% (105/158 circulated surveys). Three questionnaires were incomplete and thus excluded from analysis, resulting in a final sample of 102.

### 3.1. Demographics

Responding physicians reported a median work experience in neonatology of 16 (3–35) years. Units were admitting 5–24 (52.0%), 25–50 (39.2%), and >50 (8.8%) preterm infants annually with a birth weight <1000 g.

### 3.2. Enteral Nutrition in Preterm Infants with a Birthweight <1000 g

The majority of responders initiated enteral nutrition on the first day of life (98.0%) with colostrum (97.1%), independent of meconium passage (95.1%). Infants were exposed to breast milk (94.1%), preterm formula (52.0%), and donated breast milk (35.3%). Less frequently, glucose and maltodextrin were used (15.7% and 5.9%, respectively). Three units each used extensively hydrolyzed formula and amino acid-based formula. Full enteral feeds, i.e., enteral intake ≥140–150 mL/kg, were usually reached at median postnatal day 10 (5–21).

### 3.3. Gastrointestinal Complications

The overall approximate median annual number of NEC cases (1.0; 0–6.0), focal intestinal perforation (1.0; 0–10.0), and meconium ileus (1.0; 0–15.0) was reported to be low.

### 3.4. Use of Steroids during the First Two Weeks and Pharmacological Treatment of Patent Ductus Arteriosus

While most units did not routinely use systemic steroids during the first two weeks (63.7%), 33.3% of responders used hydrocortisone to treat arterial hypotension, and 18.6% used prophylactic low-dose hydrocortisone for bronchopulmonary dysplasia prevention. Patent ductus arteriosus (PDA) was treated using ibuprofen, paracetamol, and indomethacin (91.2%, 35.3%, and 32.4%, respectively). 6.9% of the responding units refrained from pharmacological interventions for PDA.

### 3.5. Interventions to Promote Meconium Evacuation

53.9% of neonatologists reported that they used interventions routinely to support meconium passage, 43.1% frequently, and 2.9% used such interventions only in exceptional circumstances. Details on the use of enemas, orally applied contrast agent, polyethylene glycol (PEG), and maltodextrin are listed in Table 1 and Table 2.

Nearly all responding units utilized enemas to promote meconium passage, mostly as a therapeutic intervention and less than a third as a prophylactic intervention. 92.5% (62/67) of NICUs ever administering enemas with contrast agents did so infrequently and only in exceptional cases. We found no difference in reported NEC incidence between units using either low or high osmolar contrast agents as enema agents (Figure 1; median NEC rate 1.0; 0–6.0 vs. 1.0; 0–3.0; *p* = 0.31). Furthermore, there was no difference in NEC rates for units using enemas or suppositories with or without glycerin (median NEC rate 1.0; 0–6.0 vs. 1.0; 0–3.0; *p* = 0.18). Glycerin suppositories were used by 11.0% of respondents. The number of administered enemas did not affect time until full enteral feeds. NICUs that administered two or fewer enemas per day as well as NICUs that administered more than two enemas per day reached full enteral feeds on median postnatal day 10 (*p* = 0.62).

Enemas were administered using gastric tubes (62.0%), special syringe attachments and urinary catheters (25.0% and 22.0%, respectively), rectal tubes (20.0%), and special rectal catheters for preterm infants (16.0%). One unit used endotracheal tubes due to their softness and patency. The applied volume was usually based on infant weight and varied between 2.0 and 25.0 mL/kg.

More than half the respondents used orally applied contrast agents predominantly as a therapeutic intervention. Again, we found no difference in reported NEC incidence between units that used low or high osmolar contrast agents (Figure 1; median NEC rate 1.0; 0–6.0 vs. 1.0; 0–3.0; *p* = 0.18).

While several NICUs used PEG, the use of maltodextrin was rare. Reported NEC rates did not differ between NICUs who did or did not use the aforementioned interventions (Figure 2; PEG: median NEC rate 1.0; 0.0–4.0; *p* = 0.35, Maltodextrin: median NEC rate 1.0; 0.0–1.0; *p* = 0.06). Other interventions for mobilization of meconium mentioned by respondents included abdominal massage (61.8%), rectal stimulation (52.9%), oral administration of acetylcysteine (19.6%), reflexology (1.9%), physical therapy (1.0%), and enteral administration of Tween (1.0%).

## 4. Discussion

This national survey investigated interventions used in German NICUs to promote meconium passage in preterm infants. All responders used interventions to promote meconium passage, with more than half doing so on a routine basis. Enemas of variable composition and volume constituted the most commonly used intervention, usually initiated within the first three postnatal days and administered multiple times. Although most NICUs using enemas reported doing so as a therapeutic intervention, the early initiation of enemas (median, on postnatal day 2) suggests that many NICUs did not wait for spontaneous meconium passage in preterm infants, which is known to be significantly delayed compared to term infants [1]. Saline, glycerin, mixtures thereof, and contrast agents were most commonly used as enema solutions, which is consistent with the literature [10,12,15]. In our survey, most NICUs used contrast agents as enema solutions only in refractory cases of perceived meconium obstruction. While older studies detected a nonsignificant trend toward an increased risk of NEC using glycerin enemas or suppositories [16], an updated meta-analysis did not confirm this finding [12]. In line with this, we also found no difference in NEC rates between NICUs using glycerol-containing enemas and suppositories or not.

The use of orally applied contrast agents was the second most common used method, however usually in a single, rarely repeated, therapeutic or diagnostic procedure when other measures to mobilize meconium had failed. In both cases where a contrast agent was used either as an enema solution or orally, over one third of NICUs reported administering diluted high osmolar contrast agents. This approach is understandable as these agents may soften meconium through water influx into the intestinal lumen and have been used for many years to treat meconium ileus [17,18]. On the other hand, Haiden et al. observed a higher proportion of NEC cases (8% in controls vs. 21% in intervention group) in very low birthweight preterm infants receiving a diluted high osmolar contrast agent in the first 24 postnatal hours. According to the responders to our survey, high osmolar contrast agents were usually used beyond the first 24 h of life. Again, we did not find differences in reported NEC rates between users of high vs. low osmolar contrast agents.

The interventions ranked third and fourth for promoting meconium passage were the oral administration of PEG and acetylcysteine. While PEG, an osmotic laxative, is an established and safe therapy for constipation in infancy [19,20], little or no data exist on its use in preterm infants. A possible effect on meconium passage is conceivable due to an osmotic effect similar to contrast agents. PEG has also been used in children <2 years of age without relevant side effects [21]. However, it is unclear whether the lack of relevant side effects can be transferred to the immature intestine of premature infants. Also, the dose and frequency of administration should be adapted for preterm infants. Studies on the use of PEG for meconium mobilization in preterm infants are urgently needed. 

The use of acetylcysteine to treat neonatal meconium obstruction has long been reported [22]. Acetylcysteine is applied in cystic fibrosis patients with intestinal obstruction syndrome due to its mucolytic effect, which is expected to reduce mucous viscosity, thereby facilitating meconium passage [23]. However, like PEG, little to no data exist regarding dosing and safety for preterm infants. This is also the case for maltodextrin, used in a small number of NICUs.

Regarding physical interventions, abdominal massage and rectal stimulation were reported by more than half the respondents. Both interventions are used quite commonly, but we did not obtain more specific information on their use in our survey. For abdominal massage, an intuitive measure to support peristalsis, an increased risk of intestinal volvulus without malrotation has been reported. Therefore, this intervention should be used with extreme caution [24]. While one study on rectal stimulation reports no clinically significant benefit for meconium passage [25], to our knowledge, no studies exist evaluating the effects of reflexology, physical therapy, and the enteral administration of Tween. The latter has been used as an enema solution in neonates with meconium obstruction [26].

Our study has several limitations. Despite a defined target population with whom repeated direct contact was established via e-mail, the response rate was only 64.6%. Although this may be acceptable [27], our survey provides only limited information on the interventions used to promote meconium passage in German NICUs. There is also a significant risk of reporting bias, e.g., if only NICUs with a particular focus on facilitating meconium passage participated in our survey. No responder reported watchful waiting without any intervention. Both incomplete questionnaires were terminated before section V. There may be NICUs not emphasizing early interventions to promote meconium passage and thus preferring a wait-and-see approach; this group would be underrepresented in this survey. By not enquiring about the total number of preterm infants admitted to each NICU we could not report detailed data on NEC incidence (i.e., percentages), but reported rates seem rather low. When interpreting these data, it should be considered that most NICUs admitted less than 25 infants weighing <1000 g per year. Also, inquiring about early antibiotic treatment would have been a valuable addition to our survey, as antibiotic therapy during the first postnatal days may lead to microbial dysbiosis, thus increasing NEC risk [28]. This being a survey study, we had to avoid asking too many detailed questions, which would give a more complete picture but would hardly secure a high response rate [29]. Since we provided several questions where free text input was possible, analyzing specific questions was difficult, as the topic of delayed meconium passage lacks precise terminology and definition [30].

Our survey showed that many different interventions are utilized in German NICUs to promote meconium passage in preterm infants. While more than half the NICUs used interventions on a routine basis, most rated their actions as therapeutic interventions. A significant proportion of units used enemas and orally administered PEG early after birth as a prophylactic approach, presumably to shorten the time of meconium passage and full enteral feeds. The evidence for routinely promoting meconium passage via enemas or suppositories and orally administered contrast agents is low or, especially for PEG and maltodextrin, nonexistent [6,12]. Despite insufficient data, the use of the interventions mentioned above seems appropriate in anticipation of potential surgical interventions for severe meconium obstruction. Large controlled trials are needed to evaluate efficacy and safety of routinely administered interventions to promote meconium passage in preterm infants and avoid meconium obstruction.

## 5. Conclusions

A wide variety of interventions was utilized to promote meconium passage in German NICUs. Little to no evidence exists on most of the interventions reported herein. We were unable to show any effect on NEC rates for any of the interventions reported.

## Figures and Tables

**Figure 1 children-09-01122-f001:**
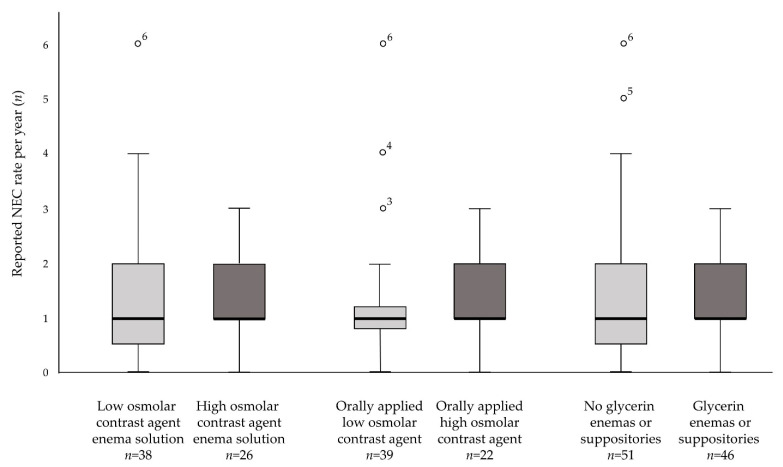
Boxplot distribution of reported NEC rate per year and interventions used (high vs. low contrast agents; use vs. no use of glycerin enemas and suppositories). Dots indicate outlier values. NEC rate was provided by 99 participating units. NEC—necrotizing enterocolitis.

**Figure 2 children-09-01122-f002:**
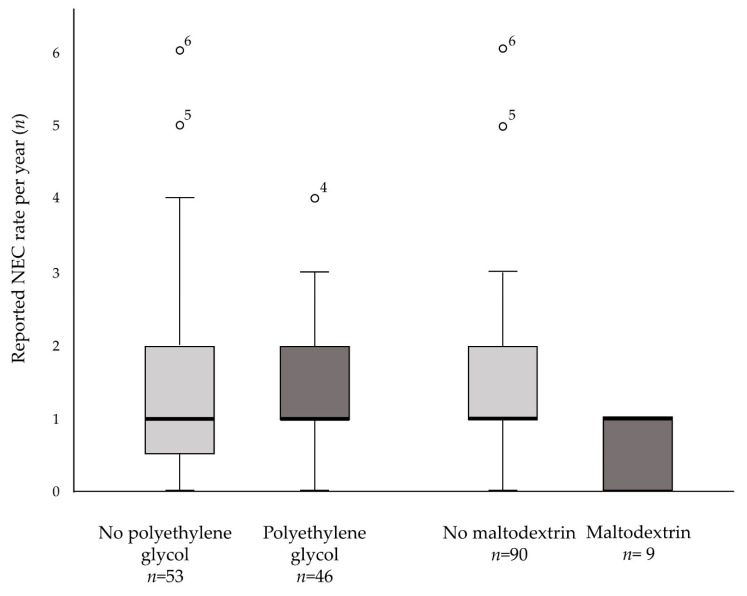
Boxplot distribution of reported NEC rate per year and interventions used (use vs. no use of polyethylene glycol and maltodextrin). Dots indicate outlier values. The NEC rate was provided by 99 participating units. NEC—necrotizing enterocolitis.

**Table 1 children-09-01122-t001:** Use of enemas and orally applied contrast agents to promote meconium passage.

Total (*n* = 102)	Enemas *n* (%)	Orally Applied Contrast Agent *n* (%)
**Use**		
Yes	100 (98.0%)	63 (61.8%)
No	2 (2.0%)	39 (38.2%)
**Intended use**	*n* = 100 #	*n* = 61 #
Prophylactic	29 (29.0%)	5 (8.2%)
Therapeutic	71 (71.0%)	56 (91.8%)
**Target population**	*n* = 100 #	*n* = 63
All preterm infants	27 (27.0%)	18 (28.6%)
<32 weeks of gestation	19 (19.0%)	6 (9.5%)
<28 weeks of gestation	27 (27.0%)	16 (25.4%)
<1500 g birth weight	32 (32.0%)	7 (11.1%)
<1000 g birth weight	26 (26.0%)	15 (23.8%)
Other criteria	5 (5.0%) “Absent passage of meconium”	7 (11.1%) “If other interventions failed”
	3 (3.0%) “No meconium until day three”	3 (4.8%) “Only in rare cases”
	2 (2.0%) “Small for gestational age”	3 (4.8%) “Small for gestational age”
	2 (2.0%) “Special cases”	3 (4.8%) “Ileus or mechanical obstruction”
		2 (3.2%) “Birth weight >1000–1500 g”
**Initiation**	*n* = 100 #	*n* = 63
First postnatal day	18 (18.0%)	3 (4.8%)
Given postnatal day	36 (36.0%) Postnatal day 2.0 (2.0–4.0)	10 (15.9%) Postnatal day 3.5 (3.0–5.0)
No or little meconium passed until	52 (52.0%) Postnatal day 3.0 (2.0–6.0)	46 (73.0%) Postnatal day 5.0 (3.0–10.0)
**Frequency**	*n* = 100 #	*n* = 59 §
Once a day	23 (23.0%)	45 (71.4%)
Multiple times a day	41 (41.0%)	10 (15.9%)
Only once or twice in total	34 (34.0%)	N/A
Based on indication	6 (6.0%) “Twice a day”	4 (6.5%) “Only once”
	4 (4.0%) “Special cases”	4 (6.5%) “Meconium ileus”
	2 (2.0%) “Three times per day”	1 (1.6%) “Two times”
	2 (2.0%) “Every 48 h”	1 (1.6%) “Three times every 48 h”
		1 (1.6%) “Meconium plugging”
**Duration of use**	*n* = 100 #	N/A
Until passing meconium at least once	35 (35.0%)	
Until passing transitional stool	45 (45.0%)	
Until passing milk stool	3 (3.0%)	
Until full enteral feeds	2 (2.0%)	
Others	15 (15.0%) “One or two spontaneous	
	bowel movements per day”	
	9 (9.0%) “Based on individual decisions”	
**Agents used**	*n* = 100 #	N/A
Normal saline	76 (76.0%)	
Contrast agent	67 (67.0%)	
Glycerin	41 (41.0%)	
Acetylcysteine	23 (23.0%)	
Glucose 5%	22 (22.0%)	
Lipid solution	9 (9.0%)	
Breast milk	5 (5.0%)	
Others	2 (2.0%) “Glucose and glycerin”	
	2 (2.0%) “Tween 0.5%”	
	1 (1.0%) “Glucose and acetylcysteine”	
	1 (1.0%) “Glucose 10%”	
	1 (1.0%) “Glycerin and distilled water”	
	1 (1.0%) “Normal saline and glycerin”	
	1 (1.0%) “Normal saline and acetylcysteine”	
	1 (1.0%) “Ringer’s solution and PEG”	
**Type of contrast agent**	*n* = 67	*n* = 63
Low osmolar	40 (59.7%)	39 (61.9%)
High osmolar	24 (35.8%)	22 (34.9%)
Both low and high osmolar	3 (4.5%)	2 (3.2%)

# Not specified *n* = 2; § Not specified *n* = 4; Data presented as total response frequency with percentages in parenthesis. Postnatal day shown as median and minimum to maximum in paracentesis. PEG—polyethylene glycol.

**Table 2 children-09-01122-t002:** Use of orally applied polyethylene glycol and maltodextrin to promote meconium passage.

Total (*n* = 102)	Polyethylene Glycol *n* (%)	Maltodextrin *n* (%)
**Use**		
Yes	47 (46.1%)	9 (8.8%)
No	55 (53.9%)	93 (91.2%)
**Intended use**	*n* = 47	N/A
Prophylactic	16 (34.0%)	
Therapeutic	31 (66.0%)	
**Target population**	*n* = 47	*n* = 9
All preterm infants	11 (23.4%)	1 (11.1%)
<32 weeks of gestation	11 (23.4%)	2 (22.2%)
<28 weeks of gestation	15 (31.9%)	2 (22.2%)
<1500 g birth weight	14 (29.8%)	0 (0.0%)
<1000 g birth weight	11 (23.4%)	2 (22.2%)
Other criteria	4 (8.5%) “Small for gestational age”	1 (11.1%) “Small for gestational age”
	3 (6.4%) “If other interventions failed”	1 (11.1%) “Impaired intestinal motility”
	2 (4.3%) “Only in rare cases”	
	1 (2.1%) “Meconium plugging”	
**Initiation**	*n* = 47	*n* = 9
First postnatal day	8 (17.0%)	6 (66.7%)
Given postnatal day	14 (29.8%) Postnatal day 3.0 (1.0–8.0)	1 (11.1%) Postnatal day 3.0 (3.0–3.0)
No or little meconium passed until	25 (53.2%) Postnatal day 3.0 (2.0–14.0)	2 (22.2%) Postnatal day 4.0 (3.0–5.0)
**Duration of use**	*n* = 47	*n* = 9
Until passing meconium at least once	12 (25.5%)	2 (22.2%)
Until passing transitional stool	18 (38.3%)	5 (55.6%)
Until passing milk stool	4 (8.5%)	0 (0.0%)
Until full enteral feeds	9 (19.1%)	0 (0.0%)
Others	3 (6.4%) “Multiple bowel movements”	1 (11.1%) “First two feeds with maltodex-
	1 (2.1%) “14 days”	trin, then one feed consisting of a 1:1 mix-
	1 (2.1%) “28 days”	ture of maltodextrin and milk
	1 (2.1%) “42 days”	or formula”

Data presented as total response frequency with percentages in parenthesis. Postnatal day shown as median and minimum to maximum in paracentesis.

## Data Availability

The data presented in this study are available on request from the corresponding author.

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
