# Peer review of "Interventions for Promoting Meconium Passage in Very Preterm Infants—A Survey of Current Practice at Tertiary Neonatal Centers in Germany"

_children, 2022, doi:10.3390/children9081122_

Round 1
Reviewer 1 Report
The Authors conducted a national survey to assess current approaches to meconium evacuation in very preterm infants and their potential association with NEC.
To my opinion the manuscript is very nice, well written, well performed and the limits are clairley discussed.
I have not important comments before the publication, only a suggestion to improve more the presentation of the results.
I suggest to performe graphics for the NEC incidence and the interventions reported. This is an interesting data of your survey, and to my opinion despite the limitations declared and discussed, should be enphatized.
Congratulations!
Reviewer 2 Report
In this study, authors aim to study the co-relation of different meconium evacuation approaches and NEC. Based on their survey, authors did not find any signifcant co-reation of any of the approcahes with NEC. This data although negative, is suggestive that most of the approaches for meconium passage doesnt predispose infants to NEC. There are some points that authors need to address as listed below.
1. In study design, please explicilty mention that the inclusion and exclusion criteria.
2. NEC has signifcant co-relation with the antibiotics given to the infants. If possible, please include the information about the antibiotics given to these infants.
